# Gastric Mobility and Gastrointestinal Hormones in Older Patients with Sarcopenia

**DOI:** 10.3390/nu14091897

**Published:** 2022-04-30

**Authors:** Hsien-Hao Huang, Tse-Yao Wang, Shan-Fan Yao, Pei-Ying Lin, Julia Chia-Yu Chang, Li-Ning Peng, Liang-Kung Chen, David Hung-Tsang Yen

**Affiliations:** 1Department of Emergency Medicine, Taipei Veterans General Hospital, Taipei 112201, Taiwan; hhhuang@vghtpe.gov.tw (H.-H.H.); tseyao85@gmail.com (T.-Y.W.); pylin13@vghtpe.gov.tw (P.-Y.L.); cychang@vghtpe.gov.tw (J.C.-Y.C.); 2Institute of Emergency and Critical Medicine, School of Medicine, National Yang Ming Chiao Tung University, Taipei 112304, Taiwan; 3Faculty of Medicine, National Yang Ming Chiao Tung University, Taipei 112304, Taiwan; 4Department of Nuclear Medicine, Taipei Veterans General Hospital, Taipei 112201, Taiwan; sfyao@vghtpe.gov.tw; 5Aging and Health Research Center, National Yang Ming Chiao Tung University, Taipei 112304, Taiwan; lining.peng@gmail.com (L.-N.P.); lkchen2@vghtpe.gov.tw (L.-K.C.); 6Center for Geriatrics and Gerontology, Taipei Veterans General Hospital, Taipei 112201, Taiwan; 7Taipei Municipal Gan-Dau Hospital, Taipei 112020, Taiwan; 8Department of Emergency Medicine, National Defense Medical Center, Taipei 114202, Taiwan; 9Department of Nursing, Yuanpei University of Medical Technology, Hsinchu 300102, Taiwan

**Keywords:** sarcopenia, gastrointestinal hormones, gastric emptying, older patients

## Abstract

Sarcopenia has serious clinical consequences and poses a major threat to older people. Gastrointestinal environmental factors are believed to be the main cause. The aim of this study was to describe the relationship between sarcopenia and gastric mobility and to investigate the relationship between sarcopenia and the concentration of gastrointestinal hormones in older patients. Patients aged ≥ 75 years were recruited for this prospective study from August 2018 to February 2019 at the emergency department. The enrolled patients were tested for sarcopenia. Gastric emptying scintigraphy was conducted, and laboratory tests for cholecystokinin(CCK), glucagon-like peptide-1 (GLP-1), peptide YY (PYY), nesfatin, and ghrelin were performed during the fasting period. We enrolled 52 patients with mean age of 86.9 years, including 17 (32.7%) patients in the non-sarcopenia group, 17 (32.7%) patients in the pre-sarcopenia group, and 18 (34.6%) in the sarcopenia group. The mean gastric emptying half-time had no significant difference among three groups. The sarcopenia group had significantly higher fasting plasma concentrations of CCK, GLP-1, and PYY. We concluded that the older people with sarcopenia had significantly higher plasma concentrations of CCK, GLP-1, and PYY. In the elderly population, anorexigenic gastrointestinal hormones might have more important relationships with sarcopenia than orexigenic gastrointestinal hormones.

## 1. Introduction

Sarcopenia, a progressive and generalized decline in skeletal muscle mass and muscle function, has serious clinical consequences and has emerged as a challenge for public health [1,2,3,4]. Sarcopenia increases the risk of disability in activities of daily living [5] and has also been shown to increase the likelihood of adverse outcomes, such as falls, fractures [6], and even mortality [7]. The prevalence of sarcopenia varies depending on the definition of sarcopenia. According to a recent study, the prevalence of sarcopenia in older people in Asian countries ranges from 5.5% to 25.7% and increases with age [2,8]. Sarcopenia predominantly occurs in males (5.1–21.0% in males and 4.1–16.3% in females) [2]. From a health care perspective, sarcopenia represents an increasing financial burden [1,2,3,5,9]. Aged people with sarcopenia are more likely to be institutionalized and hospitalized, with a prolonged hospital stay and a higher cost of daily care compared to younger patients [10,11]. Sarcopenia has been identified as a disease and has its own ICD-10- MC diagnosis code [12].

Sarcopenia has long been considered a component of geriatric syndrome and is associated with the aging process [13,14]. The pathophysiology of sarcopenia in older people is thought to be a complicated net effect of many factors, including chronic inflammation, aging, inadequate nutrient supply, malabsorption, immobility, hormonal changes, organ dysfunction, diseases, and medications [3]. However, as the knowledge of sarcopenia has evolved in recent years, it has also been recognized as a muscle disease that can develop earlier in life [15]; thus, the contributing causes may lie beyond aging [16,17]. In recent years, additional specific mechanisms of sarcopenia have been identified, including mitochondrial dysfunction [18,19], insulin resistance [17,20], and alterations in intracellular calcium homeostasis [19,21]. These specific mechanisms provide a clearer picture of sarcopenia and shed light on sarcopenia management.

In recent years, there has been an increasing interest in the relationship between the gastrointestinal environment and muscle wasting. Numerous studies have presented and addressed the gut–microbiome–muscle axis (or gut-muscle axis) hypothesis. According to previous studies, the gastrointestinal environment influences muscle mass and its function by changing nutrition and energy metabolism, altering inflammatory regulation, increasing insulin sensitivity, and affecting mitochondrial function [22,23]. In addition, some in vitro and in vivo studies have demonstrated that altering the gut microbiota has beneficial effects on sarcopenia prognosis [22,23].

As evidence of a link between the gastrointestinal tract and sarcopenia is accumulating and given the great complexity of the gastrointestinal environment [24], we were interested in the other factors of the digestive process that might affect skeletal muscle. Previous studies have demonstrated a link between sarcopenia and delayed gastric emptying in postoperative patients [25,26]. The study of frail older people also suggests that frail individuals have accelerated postprandial gastric emptying [27]. However, the relationship between sarcopenia and gastric emptying time requires further investigation. In addition, some gastrointestinal hormones are known to play a role in gastrointestinal motility [28]. A previous study also suggested that some gastrointestinal hormones are involved in the process of anorexia and are associated with aging frailty [27,29]. However, the relationship between sarcopenia and the levels of gastrointestinal hormones remains to be investigated. Developing a better understanding of the relationship between sarcopenia, gut motility, and gastrointestinal hormone changes in older people could help us gain more information in ageing process. Furthermore, some gastrointestinal hormones may become a potential intervention target when treating older sarcopenia patients in the future.

Therefore, the aim of our study was to describe the relationship between sarcopenia and gastric motility in older patients and to investigate the relationship between sarcopenia and the concentration of plasma gastrointestinal hormones that affect gastrointestinal motility in older patients.

## 2. Materials and Methods

### 2.1. Participants

This prospective study was conducted at Taipei Veterans General Hospital from August 2019 to November 2020. The included patients were all from the emergency department observation room, which has 83,000 emergency department visits annually. We included patients aged 75 years and older who were admitted to the emergency department observation room and were awaiting further laboratory testing, imaging, or admission. Patients with the following characteristics were excluded from the study: (1) unstable clinical conditions, such as unstable hemodynamic status, high oxygen demand, or pending surgical or medical procedures; (2) diagnosis of malignant disease and under tumor-related treatment or palliative care; (3) immunosuppressed state, including taking immunosuppressants; (4) inability to perform physiological tests or cooperate with history taking, blood test, or gastric emptying test; and (5) unwilling to participate in this study or unwilling to provide written informed consent. The study protocol was approved by the hospital’s Institutional Review Board (approval number: 2018-03-011CC).

### 2.2. Experimental Design

In this study, we performed a comprehensive geriatric assessment (CGA) in all included patients based on the STEP assessment [30,31]. The CGA included standard question tests on baseline data, age, education level, marital status, social status, housing situation, medical history, medication safety assessment, and review of physical functioning (hearing and visual impairment, insomnia, incontinence, mobility problems, and history of falls).

Other demographic characteristics were collected by trained research staff, including body mass index (BMI), modified cumulative illness rating scale for geriatrics and Charlson’s comorbidity index (CCI) [32] for illness history, polypharmacy (defined as current use of >4 medications over 2 weeks), cognitive impairment (defined as scores < 24, using the Chinese version of the Mini-Mental State Examination [MMSE]) [12], activities of daily living (ADLs, assessed using the Barthel Index) [33]), instrumental activities of daily living (IADL), depressive symptoms (defined as scores ≥ 2 on the 5-item Chinese Geriatric Depression Scale [34]), and malnutrition (defined as scores of 12 on the Mini-Nutritional Assessment-Short Form (MNA-SF)) [35]. Frailty was evaluated using a validated scoring system [36]. Frailty was defined as scores of 3 out of the following five domains: shrinkage (body weight loss), weakness (decreased grip strength: males, 26 kg; females, 18 kg), fatigue (effort and motivation), low physical activity (inquiring about leisure activities), and slowed walking speed.

### 2.3. Sarcopenia

In this study, patients were divided into three groups according to the definition of the European Working Group on Sarcopenia in Older People (EWGSOP): the sarcopenia, pre-sarcopenia, and normal group [37]. Patients with low muscle mass who did not pass a muscle strength test or a physical performance test were classified as having sarcopenia, and the other patients with low muscle mass were assigned to the pre-sarcopenia group.

In this study, muscle mass was measured using bioelectrical impedance analysis (BIA) [37]. The skeletal muscle index (SMI), defined as appendicular skeletal muscle mass/height^2^ (kg/m^2^), was measured by trained nurses. The diagnostic cut-off value for “low muscle mass” is <7.0 kg/m^2^ and 5.7 kg/m^2^ in males and females, respectively [2]. We conducted bioelectrical impedance analysis with a body composition analyzer InBody S10 (BIOSPACE Co., Ltd., South Korea). In our study, enrolled patients completed BIA between 8:00 and 9:00 a.m. after a fasting period since last midnight. In BIA measurement, we applied fasting period of 8–9 h for hydration status control. For skin temperature control, we conducted BIA at same ER observation room.

Muscle strength was using a handgrip strength test. Patients’ handgrip strength was measured at a 90-degree flexion of the elbow in a seated position using a handheld dynamometer (Smedlay’s Dynamo Meter; TTM, Tokyo, Japan). We documented the top three results of the dominant hand as the patient’s handgrip strength [38]. The diagnostic cut-off values for males and females were <30 kg and 20 kg, respectively. Physical performance was assessed using the timed-up-and-go test (TUG). The TUG test consists of a series of movements including standing up from a chair, walking a distance of 3 m, turning around, walking back, and sitting down [39].

### 2.4. Gastric Emptying Test

Gastric emptying time was measured by gastric scintigraphy using established control values [40]. All patients were administered a standardized 99 mTc-SC solid meal comprising 1 mCi 99 mTc-SC radiolabeled protein, two slices of white bread, 30 g of strawberry jam, and 120 mL water. After completion of the meal, scintigraphy was performed on all of the patients. The image was obtained in the anterior and posterior positions using a scintillation gamma camera with 1 min static images. The regions of interest were imaged around the stomach, and the geometric mean of the anterior and posterior views was determined. The percent retention or gastric emptying was calculated at each imaging time point. 

According to the standardized solid meal criteria [41], patients with >10% retention after 4 h were classified as having delayed gastric emptying.

### 2.5. Laboratory Test

Laboratory tests for gastric hormones were performed for all patients in this study between 7:00 and 8:00 a.m. after a fasting period since last midnight by venous blood sampling from the antecubital vein. Plasma levels of gastrointestinal hormones, including cholecystokinin (CCK), glucagon-like peptide-1 (GLP-1), peptide YY (PYY), and ghrelin, were measured using the RayBio Human ELISA kit (Ray Biotech, Norcross, GA, USA). Nesfatin levels were measured using a commercial Nesfatin-1 (1–82) (Human) ELISA Kit (Phoenix Pharmaceuticals, Belmont, CA, USA). All gastrointestinal hormones were measured using commercially available and validated analytical kits.

### 2.6. Statistical Analysis

All continuous variables in this study were presented as mean ± standard deviation, and categorical data are expressed as numbers (percentages). We used a one-way ANOVA to compare continuous variables between the groups and a chi-square test to compare categorical variables between the groups as appropriate. Variables with *p*-value < 0.1 were analyzed stepwise backward using multiple logistic regression to determine the independent predictor. Correlations between two variables were analyzed using the Pearson’s correlation method. Statistical significance was set as *p* < 0.05. SPSS for Windows (version 21.0; IBM Corp., Armonk, NY, USA) was used for all statistical analyses.

## 3. Results

### 3.1. Baseline Demographic Characteristics

All 290 patients were initially recruited, as shown in Figure 1. A total of 52 patients were finally included in this study according to the exclusion criteria mentioned in the Materials and Methods section. The mean age of the included patients was 86.9 ± 7.5 years (Table 1).

According to the definition of sarcopenia and pre-sarcopenia as mentioned above, the patients were divided into three groups: 17 (32.7%) in the non-sarcopenia group, 17 (32.7%) in the pre-sarcopenia group, and 18 (34.6%) in the sarcopenia group (Table 1).

The age of the patients was significantly higher in the sarcopenia group than in the other pre-sarcopenia and sarcopenia group (*p* < 0.05). The weight and BMI of patients in the pre-sarcopenia and sarcopenia groups were significantly lower than those in the non-sarcopenia group. The arm circumference and calf circumference were significantly lower in the sarcopenia group than in the non-sarcopenia group. The IADL and MMSE scores were significantly lower in the sarcopenia group than in the other group. The sex, height, body fat percentage, waist circumference, CCI score, MNA-SF outcomes, frailty, and number of falls in the past year showed no significant differences between the groups.

### 3.2. Gastric Empty Time between Groups

As shown in Table 2, three patients met the criteria for delayed gastric emptying; each group had one patient. The mean gastric emptying half time (GE T_1/2_) was 105.8, 99.6, and 88.9 min in the non-sarcopenia, pre-sarcopenia, and sarcopenia groups, respectively. The results of gastric emptying half-time did not show a statistically significant difference among three groups. Although in the first hour, the percentage of gastric emptying in the sarcopenia group was higher than that in the other groups, the difference was not statistically significant.

### 3.3. The Analysis of Gastrointestinal Hormone Levels between Groups

Table 3 shows the laboratory results for the concentration of gastrointestinal hormones during fasting. The mean CCK concentrations were 725.2, 385.4, and 1543.6 (pg/mL) in the non-sarcopenia, pre-sarcopenia, and sarcopenia group, respectively. The sarcopenia group had significantly higher fasting plasma CCK concentrations than both the non-sarcopenia and pre-sarcopenia groups.

The mean GLP-1 concentrations were 835.2, 678.4, and 1125.7 pg/mL in the non-sarcopenia, pre-sarcopenia, and sarcopenia groups, respectively. Compared to the non-sarcopenia and the pre-sarcopenia groups, the sarcopenia group had significantly higher fasting GLP-1 concentrations.

The mean nesfatin concentrations were 0.3, 0.3, and 1.1 ng/mL in the non-sarcopenia, pre-sarcopenia, and sarcopenia groups, respectively. The PYY concentration was significantly higher in the sarcopenia group than in the non-sarcopenia and pre-sarcopenia groups.

The mean ghrelin concentrations were 24.1, 26.3, and 24.7 ng/mL in the non-sarcopenia, pre-sarcopenia, and sarcopenia groups, respectively. The fasting plasma ghrelin levels were not significantly different between the three groups.

## 4. Discussion

In this ER-based study in elderly patients, we found no significant association between gastric emptying time and sarcopenia. In addition, we found older people with sarcopenia had significantly higher plasma concentrations of anorexigenic gastrointestinal hormones, including CCK, GLP-1, and PYY.

Sarcopenia is a muscle disease that has considerable physiological and clinical impacts. The pathophysiology of sarcopenia is multifactorial and complex, and it is even more difficult to analyze the specific causality in older people. As the understanding of sarcopenia has increased in recent years, it is believed that the gastrointestinal environment is an important component of the pathophysiology of sarcopenia. Previous studies have hypothesized that anorexia is one of the major causes of frailty and sarcopenia and that both decreased in gastric compliance and alterations in gastrointestinal hormones are the driving factors of anorexia [29,42,43]. A recent study observed that gut microbiota differences affect appetite and therefore have an impact on muscle strength [44]. However, there are insufficient data to implicate actual changes in gut hormones and motility in older people with sarcopenia.

The relationship between gastric motility and aging has been previously described, and slower gastric emptying has been noted in older people [42,45]. Some studies have shown an association between gastric motility and obesity. In obesity studies, gastric emptying rates were found to be similar in obese and normal-weight adults [46,47]. However, some studies have shown that gastric dysmotility could lead to anorexia and therefore have a major clinical impact [29,48]. Subsequently, a significant association between sarcopenia and delayed gastric emptying was observed in patients who underwent pancreaticoduodenectomy [25,26].

In a previous study, gastric emptying time was compared between frail older people, non-frail older people, and young adults [27] and revealed that frail older people had greater early postprandial gastric emptying. This study also suggests that low antral compliance leads to an increase in fasting antral volume and acceleration of early postprandial gastric emptying, which in turn leads to frailty [27,49]. However, it is noteworthy that in this study, gastric motility was evaluated by antral area ultrasound measurement and acetaminophen absorption rate, which mainly indicated liquid gastric emptying [27,50]. The correlation between gastric emptying of liquids and gastric emptying of solids is not always reliable, especially in patients with gastroparesis, with a mean age of 41.0–44.8 years [51,52,53]. In our study of patients with a mean age of approximately 86.9 years, gastric motility was assessed by gastric emptying scintigraphy using a standardized egg-labeled meal, which mainly represented solid gastric emptying. Our results showed that the gastric emptying of solids in older people was not significantly different from that in the other groups. The sarcopenia group also had greater gastric emptying, although not significant, in the first postprandial hour, which is consistent with the results of a previous study. Faster gastric emptying may happen in early postprandial period, but this requires more evidences to prove. The discrepancy between previous literature and the current study may be due to insufficient case numbers and differences in the measurement of gastric emptying. Whether faster gastric emptying with solid food in the early postprandial period affects sarcopenia requires further evidence.

Gastrointestinal hormones are another important component of gastrointestinal regulators. Gastrointestinal hormones stimulate the gastric vagal motor circuit and influence intestinal motility and satiety, thus causing anorexigenic and orexigenic effects [28,54]. Gastrointestinal hormones can be divided into two groups: an anorexigenic group, which includes cholecystokinin (CCK), peptide tyrosine (PYY), glucagon-like peptide-1 (GLP-1), and pancreatic polypeptide (PP), and an orexigenic group, which includes ghrelin and motilin. The aging process itself has an impact on gastrointestinal hormones. Indeed, it has been suggested that CCK levels increase with age and have a great effect on satiety in the elderly [49,55]. However, previous studies have not shown conclusive changes in other gastrointestinal hormones in older people (mean age, 71.2 years) [29,56]. A recent meta-analysis demonstrated the association between gastrointestinal hormones and aging. Comparing old healthy adults and young adults, the concentrations of leptin, CCK, and PYY were found to be higher in older people (mean age, 73 years), while the concentrations of ghrelin and GLP-1 showed no significant difference [57].

CCK is released from the neuroendocrine cells of the duodenum in response to dietary protein and fatty acids. Although CCK cannot cross the blood–brain barrier, it has multiple effects, primarily via vagal signaling, including reduction of gastric motility, inhibition of gastrin secretion, regulation of gallbladder contraction, and reduction of gastric acid secretion. Therefore, CCK acts as a gastrointestinal brake hormone. There is sufficient evidence to suggest that aging alters the CCK concentration [28,29,54,55,56,57]. In a study of older people with frailty, it was observed that frail older people have slightly higher peak postprandial CCK compared to non-frail older people [27].

GLP-1 is secreted by intestinal L-cells, located mainly in the distal ileum and colon, and is released primarily during the postprandial period. The most notable effects of GLP-1 are stimulation of insulin and increased glucose uptake into tissues. However, GLP-1 also has other regulatory effects on the digestive process, including enhanced satiety, delayed gastrointestinal motility, and increased acid secretion [28,29,54]. In a previous study, GLP-1 did not conclusively increase with age [56,57]. However, there is a trend, albeit insignificant, of increased GLP-1 levels in the frail older people compared to healthy older people [27]. GLP-1 also affects muscle, and a recent study demonstrated that GLP-1R agonists have therapeutic effects on muscle wasting [58].

PYY is released by L cells of the gastrointestinal tract, mainly in the ileum and colon, upon stimulation by intraluminal nutrients. The release of PYY is also regulated by other hormones, including CCK and GLP-1. PYY delays gastric emptying, inhibits intestinal motility, and suppresses the pancreas [28,54].

Increases in CCK, GLP-1, and PYY are thought to be closely related to anorexia and therefore affect muscle [29,54,57,59], and this relationship has also observed in sarcopenia patients with liver cirrhosis [60]. CCK, GLP-1, and PYY in our older people (mean age, 86.9 years) with sarcopenia had significantly higher serum levels in the three groups. We propose that anorexigenic gastrointestinal hormones, such as CCK, GLP-1, and PYY, might have important relationships with sarcopenia in the older people.

In contrast, ghrelin stimulates appetite and accelerates gastric emptying. Ghrelin is produced mainly in the stomach during fasting and acts both peripherally and centrally. Ghrelin directly triggers appetite and is involved in communication between the stomach and the brain [61]. Ghrelin not only affects digestion but also stimulates the secretion of growth hormones, affects metabolism, and is associated with aging [62]. Some studies hypothesized that ghrelin plays an important role in the maintenance of aging muscles [63,64], and found evidence in animal studies [65]. Aging individuals are generally thought to have less ghrelin than young adults; however, recent studies have suggested that ghrelin levels do not change significantly in older people [57,66]. The changes of serum ghrelin levels have not been conclusively determined [63,66]. In our study, the older people showed no significant differences in gastric emptying and ghrelin serum levels among the three groups. There was no sufficient evidence regarding to this topic previously. Therefore, we could only guess orexigenic gastrointestinal hormones, such as ghrelin, are less associated with sarcopenia in elderly. However, the levels of gastrointestinal hormones fluctuate and vary from postprandial to fasting. Further studies that include gastrointestinal hormones analysis both in fasting and postprandial phase should be conducted to find more evidences.

To the best of our knowledge, our study is the first to demonstrate a difference in the concentration of gastrointestinal hormones between older people with and without sarcopenia. Elevated hormones, including CCK, GLP-1, and PYY, act as important gastrointestinal inhibitory hormones. However, no significant delay in gastric emptying time was found in older patients with sarcopenia. It is noteworthy that ghrelin did not show a significant change among the groups in our study. Gastrointestinal inhibitory hormone levels were significantly increased in older people with sarcopenia. Our study successfully demonstrated the association between gastrointestinal inhibitory hormones and sarcopenia in the older population. However, the pathophysiology and possible interventions need to be substantiated by future studies.

There are some limitations to our study. First, the uncertainty, heterogeneity, and diversity of medications in our older patients may have affected the results. Second, veteran patients represented the largest proportion of the study cohort. Therefore, sex, socioeconomic status, and other confounders were the limiting factors. Third, we collected the data before the updated publication of the diagnosis of sarcopenia [2,3], and the diagnosis of sarcopenia follows the old consensus [37].

## 5. Conclusions

We conclude that the older people in the sarcopenia group had significantly higher plasma concentrations of CCK, GLP-1, and PYY. In the older population, anorexigenic gastrointestinal hormones might have more important relationships with sarcopenia than orexigenic gastrointestinal hormones.

## Figures and Tables

**Figure 1 nutrients-14-01897-f001:**
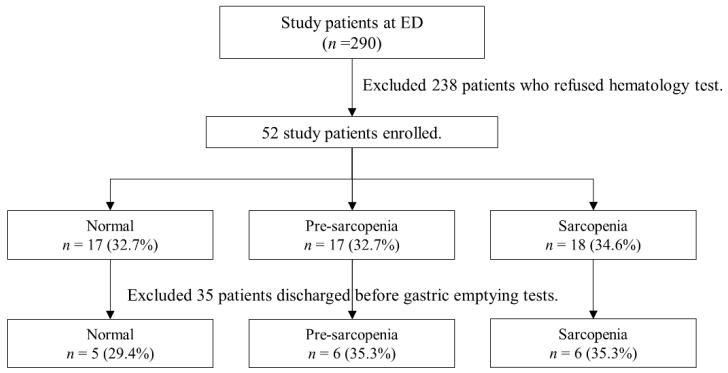
Flow diagram of the study.

**Table 1 nutrients-14-01897-t001:** Baseline demographic characteristics, variables of clinical assessment, and comprehensive geriatric assessment of older patients between groups.

	All (*n* = 52)	Non-Sarcopenia ^@^ (*n* = 17)	Pre-Sarcopenia ^#^ (*n* = 17)	Sarcopenia (*n* = 18)	*p*-Value
Age, y	86.9 ± 7.5	85.4 ± 8.1	84.6 ± 8.2	90.3 ± 4.9	0.046 *
Sex					0.388
Male	44 (84.6)	16 (94.1)	14 (82.4)	14 (77.8)	
Female	8 (15.4)	1 (5.9)	3 (17.6)	4 (22.2)	
Height	160.0 ± 7.9	160.9 ± 8.3	160.8 ± 8.2	158.5 ± 7.5	0.602
Weight	58.1 ± 12.4	66.2 ± 9.7	54.5 ± 10.7 ^@^	54.0 ± 12.9 ^@^	0.003 *
BMI	22.6 ± 4.4	25.6 ± 3.9	20.9 ± 3.7 ^@^	21.3 ± 4.2 ^@^	0.001 *
Percent body fat, %	30.4 ± 9.9	34.2 ± 9.8	27.3 ± 7.2	29.7 ± 11.2	0.116
Skeletal muscle mass, kg	21.0 ± 4.3	23.3 ± 4.0	20.6 ± 4.0	19.2 ± 4.1 ^@^	0.015 *
Total body water, %	29.3 ± 5.4	32.0 ± 5.4	28.6 ±4.9	27.4 ± 4.9 ^@^	0.029 *
Soft lean mass, kg	37.4 ± 6.9	40.9 ± 6.8	36.5 ± 6.3	34.8 ± 6.4 ^@^	0.024 *
Fat free mass, kg	39.7 ± 7.2	43.3 ± 7.4	38.9 ± 6.6	37.1 ± 6.6 ^@^	0.033 *
Fat, kg	18.4 ± 8.1	22.9 ± 7.6	15.2 ± 6.1 ^@^	16.9 ± 8.6	0.013 *
Arm circumference, cm	27.9 ± 6.2	31.0 ± 8.2	26.5 ± 3.2	25.9 ± 6.2 ^@^	0.030 *
Waist circumference, cm	83.4 ± 15.1	88.7 ± 19.9	81.4 ± 9.8	79.9 ± 12.8	0.192
Calf circumference, cm	29.1 ± 3.9	31.8 ± 3.6	29.3 ± 1.6	26.8 ± 4.4 ^@^	0.007 *
Visceral fat area, cm^2^	100.0 ± 53.4	115.7 ± 69.4	80.1 ± 37.3	102.8 ± 44.8	0.153
CCI	1.9 ± 1.8	1.6 ± 1.6	1.8 ± 1.7	2.3 ± 2.0	0.596
Barthel index	77.3 ± 27.4	73.5 ± 30.6	95.3 ± 6.4 ^@^	62.5 ± 28.5 ^#^	0.001 *
IADL	4.3 ± 2.9	4.3 ± 3.3	5.9 ± 1.6	2.3 ± 2.4 ^@ #^	0.001 *
MMSE	17.6 ± 6.6	18.3 ± 6.6	20.3 ± 5.3	14.3 ± 6.8 ^#^	0.030 *
MNA-SF					0.311
Normal nutrition	15 (33.3)	7 (53.8)	5 (31.3)	3 (18.8)	
At risk of malnutrition	20 (44.4)	5 (38.5)	7 (43.8)	8 (50.0)	
Malnutrition	10 (22.2)	1 (7.7)	4 (25.0)	5 (31.3)	
Frailty	35 (67.3)	9 (52.9)	10 (58.8)	16 (89.9)	0.051
Hand grip, kg	17.6 ± 9.0	17.1 ± 9.4	22.8 ± 8.5	13.0 ± 6.5 ^#^	0.006 *
Fall in past year	13 (28.9)	2 (15.4)	4 (25.0)	7 (43.8)	0.224
Incontinence	9 (20.0)	3 (23.1)	0	6 (37.5)	0.028 *

Data are presented as mean ± standard deviation or number (%). Abbreviations: BMI, body mass index; CCI, Charlson Comorbidity Index; IADL, Instrumental Activities of Daily; MMSE, Mini-Mental State Examination; MNA-SF, Mini Nutritional Assessment-short Form. * *p* < 0.05; comparing patients with non-sarcopenia, pre-sarcopenia, and sarcopenia. ^@^
*p* < 0.05; compared with non-sarcopenia group. ^#^
*p* < 0.05; compared with pre-sarcopenia group.

**Table 2 nutrients-14-01897-t002:** Results of gastric emptying scintigraphy.

	All (*n* = 17)	Non-Sarcopenia ^@^ (*n* = 5)	Pre-Sarcopenia ^#^ (*n* = 6)	Sarcopenia (*n* = 6)	*p*-Value *
Delayed gastric emptying	3 (17.6)	1 (20.0)	1 (16.7)	1 (16.7)	0.987
Gastric emptying half time (min)	97.7 ± 42.2	105.8 ± 45.7	99.6 ± 46.7	88.9 ± 41.2	0.817
Gastric empty at HR1 (%)	36.0 ± 15.6	27.2 ± 12.5	32.2 ± 14.8	47.2 ± 13.6	0.070
Gastric empty at HR2 (%)	69.3 ± 21.0	68.2 ± 16.2	67.0 ± 27.4	72.5 ± 20.9	0.906
Gastric empty at HR3 (%)	80.9 ± 18.4	82.0 ± 22.0	78.2 ± 18.5	82.3 ± 18.3	0.909
Gastric empty at HR4 (%)	91.8 ± 8.7	95.2 ± 6.9	89.7 ± 8.5	91.0 ± 10.6	0.584

Data are presented as mean ± standard deviation. HR1, first hour; HR2, second hour; HR3, third hour; HR4, fourth hour. * *p* < 0.05; comparing patients with non-sarcopenia, pre-sarcopenia, and sarcopenia. ^@^
*p* < 0.05; compared with non-sarcopenia group. ^#^
*p* < 0.05; compared with pre-sarcopenia group.

**Table 3 nutrients-14-01897-t003:** Analysis of gastrointestinal hormones between groups.

	All (*n* = 52)	Non-Sarcopenia ^@^ (*n* = 17)	Pre-Sarcopenia ^#^ (*n* = 17)	Sarcopenia (*n* = 18)	*p*-Value
CCK _(pg/mL)_	935.0 ± 167.7	725.2 ± 112.8	485.4 ± 72.3	1543.6 ± 429.6 ^#^	0.020 *
GLP-1 _(pg/mL)_	883.3 ± 69.8	835.2 ± 100.3	674.8 ± 120.3	1125.7 ± 119.9 ^#^	0.023 *
PYY _(pg/mL)_	39.4 ± 6.4	29.4 ± 5.2	25.2 ± 4.6	61.7 ± 16.0 ^#^	0.032 *
Nesfatin _(ng/mL)_	0.6 ± 0.2	0.3 ± 0.1	0.3 ± 0.1	1.1 ± 0.6	0.263
Ghrelin _(ng/mL)_	25.0 ± 1.6	24.1 ± 2.9	26.3 ± 2.8	24.7 ± 12.2	0.858

Data are presented as mean ± standard error of the mean. Abbreviations: CCK, Cholecystokinin; GLP-1, Glucagon-like peptide-1; PYY, Peptide YY. * *p* < 0.05; comparing patients with non-sarcopenia, pre-sarcopenia, and sarcopenia. ^@^
*p* < 0.05; compared with non-sarcopenia group. ^#^
*p* < 0.05; compared with pre-sarcopenia group.

## Data Availability

The data presented in this study are available on request from the corresponding author.

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
