# Peer review of "Gastric Mobility and Gastrointestinal Hormones in Older Patients with Sarcopenia"

_nutrients, 2022, doi:10.3390/nu14091897_

Round 1

Reviewer 1 Report

The Authors focused on a study of the Gastric mobility and gastrointestinal hormones in oldest old patients with sarcopenia. This is an interesting and comprehensive study. The article is well structured.

In my opinion:

-                The abstract presents an accurate description of this article.

-                An Authors was conducted adequate literature review.

-                The references support the rationale for reporting the study.

-                The subjects are described adequately.

-                The management of the study is effectively described.

-                Valid and reliable outcome measures are utilized.

-                The management of the article is effectively described.

-                The conclusions are appropriate.

-                Overall impression about the quality of the study is good.

Key points to consider:

Please explain the abbreviations in whole paper - if first time in article.

Figure 1 is of poor quality.

In the paper, words were sometimes written in a different font, please correct one of the versions (for example line 153, 162, Table 2, 3)

Line 27 - add the abbreviation CCK - first time in article

Line 130, 131, 132 - m2 ‘2’ should be superscript, please correct in whole manuscript

Author Response

Reviewer 1#

Point-by-point responses

  1. The Authors focused on a study of the Gastric mobility and gastrointestinal hormones in oldest old patients with sarcopenia. This is an interesting and comprehensive study. The article is well structured

Response: Thank you so much for your review. We really appreciate your taking the time to share your precious experience and scope with us.

  1. Please explain the abbreviations in whole paper - if first time in article

Response: Thank you for the important reminder. We have added the abbreviations.

  1. Figure 1 is of poor quality.
    Response: Thank you for this reminder. We have adjusted Figure 1.( Section 5.3, Page 4)

  2. In the paper, words were sometimes written in a different font, please correct one of the versions (for example line 153, 162, Table 2, 3)
    Response: We have corrected the mistake. Thank you very much.  

  3. Line 27 - add the abbreviation CCK - first time in article
    Response: Thank you for the important We have added the abbreviations. (Abstract, line 7).

 Line 130, 131, 132 - m2 ‘2’ should be superscript, please correct in whole manuscript

  1. Response: We have corrected this typing mistake. Thank you for pointing this out. (Section 3, line 136-137).

Reviewer 2 Report

This paper investigated the relationship between sarcopenia and gastric mobility and gastrointestinal hormones. This paper was written in good structure and was easy to read. However, it can be improved further if further detail can be provided.

Introduction

The definition of ‘oldest old’ is not clear. The requirement and necessity to investigate the relationship between sarcopenia and the levels of gastrointestinal hormones remains to be investigated in older people and should be addressed.

Method

Further detail should be provided. For example, the facility (e.g. brand, types) of BIA should be illustrated. The procedure to control body water has not been mentioned.

Results and Discussion

Some parts seem repeatedly (e.g. 237-243). Further discussion should be added to investigate the possible reason which makes the difference between previous literature and the current study. Further suggestions in practice should also be considered to be added in the discussion sections.

Conclusion

Line 355 It seems there is a typo (‘import’ -> ‘important’).

Author Response

Reviewer 2#

  1. This paper investigated the relationship between sarcopenia and gastric mobility and gastrointestinal hormones. This paper was written in good structure and was easy to read. However, it can be improved further if further detail can be provided.

Response: Thank you so much for your review. We really appreciate your taking the time to share your precious experience and scope with us. We have adjusted the Discussion section. Thank you very much.

  1. The definition of ‘oldest old’ is not clear. The requirement and necessity to investigate the relationship between sarcopenia and the levels of gastrointestinal hormones remains to be investigated in older people and should be addressed

Response: Thank you for the comment. The word “oldest old” is a confusing word and it is not brought benefit to reading. We have replaced “oldest old” by “older” in whole paper.
Sarcopenia has long been considered a component of geriatric syndrome and increasing in older people. The etiology of sarcopenia in young sick patient might resulted from major surgery liver cirrhosis and so on.  However, the pathophysiology of sarcopenia in old people remained unclear.
It is believed gastrointestinal hormones changed when ageing and gastrointestinal hormones may involve in process of anorexia. To illustrate the relationship between sarcopenia and the levels of gastrointestinal hormones could let us gain more understanding in ageing process and some gastrointestinal hormones may become a potential intervention target when facing old sarcopenia patients. We tried to clarify the importance of our study and added some sentences. (Section 1, line 82 -85).

  1. Further detail should be provided. For example, the facility (e.g., brand, types) of BIA should be illustrated. The procedure to control body water has not been mentione
    Response: Thank you for this critical comment. It is truly essential to clearly describe the detail of bioelectrical impedance analysis (BIA) in our study. We have adjusted the Method section.
    We conducted bioelectrical impedance analysis with a body composition analyzer InBody S10 (BIOSPACE CO. Ltd, South Korea). In our study, all 52 patients completed BIA between 8:00 and 9:00 am after a fasting period since last midnight. As we known, skin temperature, the hydration status are some important variables to controls when measuring BIA. In our study, for hydration status control, we applied fasting period of 8-9 hours. For skin temperature control, we conducted BIA at same observation room.   (Section 2.3, line 137-142).

  2. Some parts seem repeatedly (e.g. 237-243). Further discussion should be added to investigate the possible reason which makes the difference between previous literature and the current study. Further suggestions in practice should also be considered to be added in the discussion sections.
    Response: Thank you for the kindly reminder. We have adjusted the first paragraph of discussion section.

There are some discrepancies between previous literature and the current study:

  1. Previous study suggested frail older people had greater early postprandial gastric emptying, but our study did not demonstrate significant increasing gastric emptying in older sarcopenia people.

The reason of this discrepancy may be resulted from: 1. different measurement (solid or liquid), 2. Faster gastric emptying may happen in early postprandial period, we did not focus on the early postprandial period.

We have adjusted the paragraph.
(Section 4, line 291-293).

  1. In our study, we demonstrated older sarcopenia patients did not have significant differences in ghrelin level. Previous literature hypothesized that ghrelin plays an important role in the maintenance of aging muscles.

There was no sufficient evidence regarding to this topic previously. Therefore, we could only guess orexigenic gastrointestinal hormones, such as ghrelin, are less associated with sarcopenia in elderly. However, the levels of gastrointestinal hormones are fluctuated, and vary from postprandial to fasting. We measured ghrelin at fasting period only. The levels of gastrointestinal hormones in postprandial phase may have significant results. Further study, which have gastrointestinal hormones analysis both in fasting and postprandial phase, should be conducted to find more evidence.

We have adjusted the paragraph.
(Section 4, line 350-354).

  1. Line 355 It seems there is a typo (‘import’ -> ‘important’).

Response: We have corrected this typing mistake. Thank you for pointing this out.  (Section 5, line 374).
